# Enhancing Bread Quality and Shelf Life via Glucose Oxidase Immobilized on Zinc Oxide Nanoparticles—A Sustainable Approach towards Food Safety

Jahangir Khan [1], Shazia Khurshid [1,*], Abid Sarwar [2], Tariq Aziz [3], Muhammad Naveed [4], Urooj Ali [4], Syeda Izma Makhdoom [4], Abad Ali Nadeem [2], Ayaz Ali Khan [5], Manal Y. Sameeh [6], Amnah A. Alharbi [7], Faten Zubair Filimban [8], Alexandru Vasile Rusu [9,10], Gülden Göksen [11] and Monica Trif [12,*]





1 Department of Chemistry, Government College University Lahore, Lahore 54000, Pakistan
2 Pakistan Council of Scientific Industrial Research (PCSIR), Lahore 54600, Pakistan
3 School of Food & Biological Engineering, Jiangsu University, Zhenjiang 212013, China
4 Department of Biotechnology, Faculty of Science and Technology, University of Central Punjab, Lahore-54590, Pakistan
5 Department of Biotechnology, University of Malakand, Chakdara 18800, Pakistan
6 Chemistry Department, Faculty of Applied Sciences, Al-Leith University College, Umm Al-Qura University, Makkah 24831, Saudi Arabia
7 Department of Biochemistry, Faculty of Science, University of Tabuk, Tabuk 71491, Saudi Arabia
8 Division of Plant Sciences, Department of Biology, King Abdulaziz University, Jeddah 21551, Saudi Arabia
9 Life Science Institute, University of Agricultural Sciences and Veterinary Medicine Cluj-Napoca, 400372 Cluj-Napoca, Romania
10 Animal Science and Biotechnology Faculty, University of Agricultural Sciences and Veterinary Medicine Cluj-Napoca, 400372 Cluj-Napoca, Romania
11 Department of Food Technology, Vocational School of Technical Sciences at Mersin Tarsus Organized Industrial Zone, Tarsus University, 33100 Mersin, Turkey
12 Centre for Innovative Process Engineering (CENTIV) GmbH, 28857 Syke, Germany
* Correspondence: shaziakhurshid1@yahoo.com (S.K.); monica_trif@hotmail.com (M.T.)

**Abstract:** The foremost wastage of bakery products which mainly disturbs the food supply chain, especially in remote areas, is fungal growth. Good quality bread, especially with good height and volume, is the demand of every customer. Here, we aimed to develop a unique antimicrobial approach for the enhancement of the quality aspects and longevity of bread, using the synthesis of hydrogen peroxide in bread, the glucose oxidase (GOx) bioactivity, and oxidation of thiol protein bonds, which greatly enhance dough rheology, volume, and height by providing structural stability to the bread. An *Aspergillus niger*-purified enzyme was immobilized on zinc oxide nanoparticles (ZnONPs) and afterwards immersed in a buffered solution to create a mixture of GOx/ZnONPs. Analyses conducted after localization revealed that the immobilized enzyme was more active than the mobilized enzyme. GOx/ZnONPs were employed in the mixing process of bread production. The treated and control groups were evaluated for dough rheology and quality metrics including bread height and volume and storage at ambient temperature and conditions to determine shelf life by demonstrating fungal growth. In addition, antimicrobial activity was evaluated by measuring the microbiological load in terms of colony-forming units. Contrary to the control, the use of GOx/ZnONPs significantly improved bread quality, particularly bread height by 34.4%, crumb color, and volume by 30%. The shelf life of bread treated with GOx/ZnONPs was greatly extended, and the microbiological load, including yeast and mold, and total bacterial count were much lower in the GOx/ZnONPs treatment group than in the control group.

**Keywords:** bread quality; fungal growth; ZnO nanoparticles; glucose oxidase; immobilization

## 1. Introduction

Health security and food quality have been the focus of continuous research with a focus on application. Numerous health advantages have been linked to the intake of baked products containing large amounts of stabilizers and conditioners. Consumption and manufacture of meals with little preparation time have increased in popularity in recent years. Throughout the world, consumers' demand is increasing for natural fresh products, especially bread [1]. So, the biggest issue for the bread industry is serving the customers with the best texture with the natural aroma of these food products throughout their shelf life. Bread is the most popular and oldest food used by humans as a major food. It is also produced and most often approved for sale as a cultured product created from water, salt, yeast, and wheat via many processes including combining, kneading, proving, molding, and baking [2,3].

The viscoelasticity of wheat flour dough is because of gluten protein, which interacts and swells due to water compatibility embedding the starch granules. A well-defined characterization of the dough needs rheological studies [4,5]. The gluten matrix that proteins create is what gives bread dough its excellent characteristics. Oxidizing sulfhydryl (SH) and SH-disulfide (SS) exchange are the main reactions that create SS crosses along with other covalent bonds. The final product's quality characteristics are attributed by this cross-linking [6]. While baking, starch is gelatinized, heat-set, and gluten protein is pasted, resulting in the classic solid foam-back bread structure [7]. To increase the qualitative features of bread, many enzymes are employed in the baking industry [8]. The most common enzyme employed in the bread baking business is glucose oxidase (B-D glucose: oxy: 1-oxireductase; EC1.1.3.4), which is derived from several fungi, mostly from *Aspergillus niger* [9,10].

One molecule of GOx has two active sites that catalyze the conversion of B-D-glucose into hydrogen peroxide ($H_2O_2$) and gluconic acid [11]. The most significant industrial use of GOx is to extend the food products' shelf life [12]. The main industrial applications of GOx are baking, wine, production of egg powder, and gluconic acid [11]. Under Food and Drug Administration (FDA) classification, it is generally recognized as safe (GRAS) [13]. Traditionally, to strengthen gluten and improve bread's final volume and texture, different oxidants are utilized by the baking industry [14], the commonest being potassium bromate ($KBrO_3$) [15]. Although, $KBrO_3$ has been banned in many countries due to its carcinogenic nature [16] and replaced mainly by a safe alternative, which is GOx [17,18]. GOx is an effective oxidant that has consistently been shown to improve the texture, height, and volume of bread [19–21].

Shelf-life extension has been achieved by GOx [22–24], as it has antimicrobial activity on a large number of fungal strains and foodborne bacteria [25]. However, due to its instability, it has limited industrial applications; therefore, going forward, a novel target is to make it stable by immobilization.

Enzyme immobilization by physical adsorption or covalent attachment makes the enzyme more stable and functional. However, physical adsorption has its own limitations, such as unwanted enzyme diminution, casing of compressed areas, and lowering of enzyme activity. On other hand, in covalent immobilization of the enzymes, open active sites are available on the particle surface of enzymes; thus, activities of the enzymes are improved [26]. Prior to this, GOx was immobilized on iron oxide nanoparticles, which led to increased thermal stability, according to research by Abbasi et al. [27]. Numerous different natural and artificial polymers, including but not limited to chitosan, alumina, starch, magnesium silicate, silica, polyesters, gelatin, and alginate, have been utilized to immobilize enzymes up to this point [28–31]. Researchers worldwide are putting increased emphasis on this material in recent years due to its simplicity in production, growth of its surface area, biocompatibility, and low toxicity [32–34]. Given this situation, research by Lee et al. [33] demonstrated that enzymes based on nanoparticles had better stability and strength in comparison to their counterparts. This was due to the longevity of the nanoparticles. The zinc oxide nanoparticles (ZnONPs) have antibacterial capabilities

as well as zinc acting as a nutritional supplement, so it is the most popular strategy for immobilizing enzymes. ZnONPs are now reported in the packaged food market as a preferred alternative. To extend the longevity of the bread, we provide an idiosyncratic antimicrobial-based strategy that targets the industrial significance of GOx hybrids. The purpose of this research was to investigate how the bioconjugate of GOx/ZnONPs affected the bread's lifespan and quality and how the bioconjugate of GOx/ZnONPs is within the tolerance limit of the human body.

## 2. Materials and Methods

### 2.1. Chemicals

Merck (Darmstadt, Germany) Germany's zinc nitrate hexahydrate, 25% glutaraldehyde, sodium hydroxide, Sigma Aldrich's L-Cysteine-HCL, 3, 5, di nitro salicylic acid (DNSA), 2, 2-diphenyl-1-picrylhydrazyl (DPPH) (glucose oxidase refined from *Aspergillus niger*, baker's yeast, sucrose, and Oxide Germany's lysine medium, Novobiocin, carbon base yeast, glycine, L-lysine monohydrochloride, Bacteriological agar No. 1.

### 2.2. Preparation of ZnO Nanoparticles and Surface Modification

Co-precipitation method was used to prepare zinc oxide nanoparticles following standard method. Briefly, in 250 mL flask, 100 mL of zinc nitrate (0.5 M) was taken, while sodium of the same strength was supplemented drop by drop. The mixture was agitated at 60° Celsius for almost two hours until the color was changed from milky to clear solution followed by centrifugation at 3000–4000 rpm for 15 min to obtain white precipitate. The resulting precipitate was repeatedly cleaned with ethanol and deionized water before being dried for 180 min at 100 °C to produce a powder mixture that is dried by pestle and mortar method. Calcination was performed at 500 °C for 2 h to obtain ZnONPs. The schematic diagram of *ZnoNPs* preparation is shown in Figure 1.

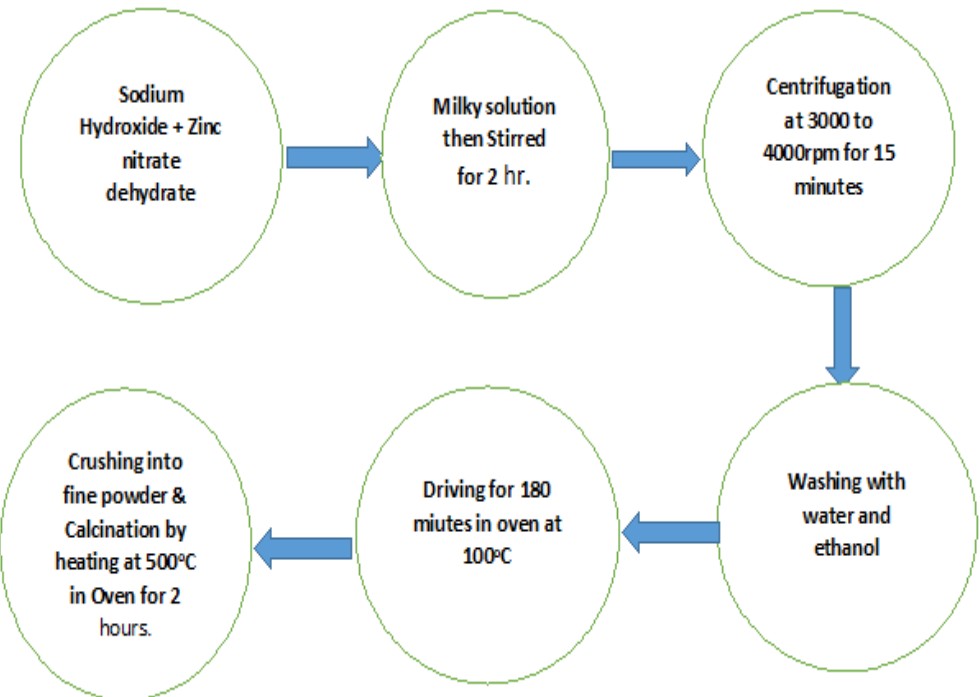

**Figure 1.** Schematic diagram of *ZnoNPs* preparation.

Surface modification of ZnONPs was performed with cysteine before going to mobilization process. The process involves adding 100 mg of ZnONPs in 20 mL of ethanol and sonication for 6 h. The mixture was allowed to dry at ambient temperature followed by addition of 20 mL of potassium phosphate buffer 0.1 M was used for suspension of

ZnONPs. An amount of 25 μL of 25% of glutaraldehyde solution was used in the mixed solution. The application's resulting product was washed after filtration [35,36].

### 2.3. ZnO Nanoparticles Characterization

Double beam UV–vis spectrophotometer (PerkinElmer Lambda 365) with wavelength of 250–750 nm was used to track the production of ZnONPs. Using an FEI Nova-Nano SEM-450 USA scanning electron microscope, the shape and size of the produced ZnONPs were examined. Drop by drop, 10 μL of ZnONPs suspension was applied to the copper grid with carbon plating. SEM samples that had been dried were carefully analyzed. The X-Ray diffractometer (XRD) confirmed that the produced materials are crystalline. Using Ca K1 irradiation with 1.5406 A wavelength, XRD examination was carried out using a BRUCKER P2 PHASER in 20 locations ranging from 0 to 80 degrees.

### 2.4. GOx Immoblization on Modified ZnONPs

*Aspergillus niger* was cultivated under standardized growth conditions of 19.5 °C and pH 5.48 to produce the GOx enzyme. GOx was immobilized by mixing 50 mg of ZnONPs with 15 mg of active enzyme in 1 mL of potassium phosphate buffer solution (0.1 M). The mixture was centrifuged for 5 min at 9000 rpm after being refrigerated at 4° C for 3 h. Divided liquid was recovered, and the settling solid was washed with buffer to dispose of the enzyme that was not bound. The supernatant and the solid combination were analyzed using UV–visible technology at a 546 nm absorption [37]. Enzyme activity of both free enzymes and supernatant was used to assess the effectiveness of GOx immobilization on ZnONP surface.

### 2.5. FTIR Analysis

FTIR was used to measure the interaction between biomolecules and nanomaterial. To prepare samples for FTIR measurement, GOx produced from Aspergillus Niger was freeze dried (lyophilized). Similarly, the GOx/ZnONPs mixture was centrifuged, cleaned, and dried before being used as an FTIR sample for an immobilized enzyme. Analyzing FTIR using Bruker's AL-PHA-P spectrometer, both GOx/ZnONPs and GOx were examined. To check for morphological alterations brought on by the adsorption of GOx on ZnONPs' exterior, GOx/ZnONPs were subjected to an SEM examination. This was accomplished by carefully drying 10 L of a colloidal mixture of GOx and ZnONPs on a copper grid covered with carbon.

### 2.6. Assessement of Enzymatic Activities

The catalytic activities of GOx (free and associated) were assessed by DNSA assay operation parameters (pH 7.0 and 25 ± 1 °C). In the presence of GOx, the alkaline 3,5 dinitro salicylic acid was reduced by glucose to form 3-amino 5-nitro salicylic acid. The reaction mixture's absorbance was measured at 546 nm with a UV–vis spectrophotometer (PerkinElmer Lambda 365). On the basis of the variation in glucose level, the catalytic performance of GOx was evaluated. Using a typical glucose profile, the reaction mixture's unknown glucose concentration was determined. The enzymatic activity was measured as:

$$\text{Enzyme Activity (U/mL)} = \frac{\text{Glucose Conc.} \times 1000}{\text{NF} \times \text{DF}}$$

NF: normalization factor.
DF: dilution factor.

### 2.7. The Bioconjugate (GOx/ZnONPs) Preparation

GOx/ZnONPs bioconjugate was dissolved in 50 mL of potassium phosphate buffer (0.1 M) to produce a suspension (pH 7). The solution was then sonicated, followed by the addition of 0.2 mL of surfactant. To evaluate the bread, the organic solvent was utilized while it was being mixed.

### 2.8. GOx/ZnONPs Experimental Design and Application of solution in Bread Making

Dough of bread was produced by combining basic ingredients—yeast, sugar, flour, and salt—along with ingredients such as GOx and GOx/ZnONPs by mechanical means into a well-developed homogeneous mass. Then, after fermentation at 25 °C to 27 °C for ten minutes, dough was divided by specific size and shape. Final molding was performed after intermediate proofing. After molding, the bread dough was allowed to relax and ferment until it reached the final volume prior to the baking process. This is carried out in a specialized chamber named a proofer with a set humidity of 80% and temperature of 37 °C for 85 min. Then, the bread was baked at 170 °C for 30 min. Bread was cooled, sliced, and then put for shelf-life study and bread height and volume measurement. Height was measured by putting a measuring scale in the center of the bread.

### 2.9. Volume of Bread Was Measured by Seed Displacement Apparatus

Seed displacement apparatus, as shown in Figure 2, was used for measurement of volume of different bread. Volume meter was opened by unlatching the lower housing and swinging it to the down position. The volume gauge was closed by returning it to its upright posture once the sample had been inserted into the bottom housing's center. Rapeseed was able to fill the closed lower housing once the bottom clasp was locked and the gate slider was released.

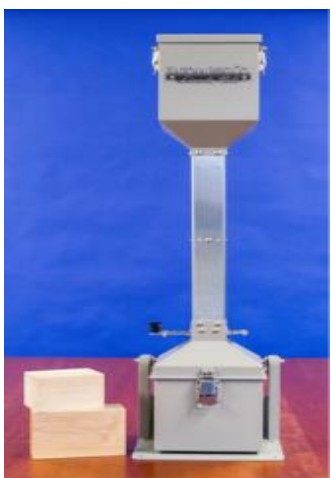

**Figure 2.** Seed displacement apparatus.

As shown by the quantity of rapeseed in the cylinder, the volume was immediately read from the standardized column. The volume record was recorded, and this action was repeated for a minimum of three samples from the same batch to calculate an average volume.

### 2.10. Enumeration of Yeast and Mold in Bread by Colony Count—GLA Method

Yeast and mold need a nitrogen source to grow. The amino acid lysine can be used by all yeast and mold strains. Therefore, using lysine medium (Oxide Germany) as a sole nitrogen source, all yeast and mold will be able to grow on the plates [38]. The same holds true for a combination of the amino acids glycine and lysine. Adding glycine to the medium will result in better growth of yeast and mold species, so it is easier to count the colonies after a shorter incubation time. Novobiocin (Merck, Darmstadt, Germany) is added to the medium to suppress the growth of any bacteria that may be present in the sample.

### 2.11. Sensory Evaluation of Bread Quality Attributes

Softness was evaluated by hand compression. Control was scored at a number 5; if the softness was worse, it was scored from 0–4; if softness was better than the control, it was scored from 6–10. The difference in texture (fineness and regularity of crumb) was evaluated with a score of 5 being the control, and other samples rated against the control

scored 0–4 for being worse than the control and 6–10 for being better than the control. Similarly, the whiteness of crumbs all samples was evaluated against the control.

### 2.12. Analytical Statistics

Using Statistix 10 program, data were statistically evaluated by ANOVA and shown as mean standard error. The least significant difference (LSD) test was used to compare several mean scores. At $p < 0.05$, variations were deemed statistically significant. Origin Pro 8 was employed to plot the data.

## 3. Findings and Analysis

### 3.1. Assay Strategy

Bread's shelf life was extended by GOx's bioactivity, and GOx's stability and metabolic activity were increased by immobilization [27]. GOx was therefore immobilized on the altered ZnONPs surface and then contained in a buffer that was used during the mixing phase of bread creation. Hydrogen peroxide and gluconic acid are produced when GOx and glucose combine. The bread's shelf life is increased, and microbiological assaults are warded off thanks to the hydrogen peroxide content.

### 3.2. ZnONPs Synthesis

White precipitates were the first sign that ZnONPs had formed. Using a UV–vis spectrometer with a wavelength range of 300–800 nm, the formulation of ZnONPs was verified. ZnONP production was verified by a sharp peak at 378 nm (Figure 3a) [39]. XRD analysis of the crystalline nature of ZnONPs revealed the hexagonal wurtzite structure (Figure 3b). These planes correspond to (202), (004), (201), (112), (200), (103), (110), (102), (101), (002), and (100). The lack of additional impurity bands in the XRD data supports the better purity of produced ZnO nanoparticles. The very strong and condensed peaks demonstrated the superb crystalline quality of the synthesized ZnONPs. Our XRD findings line up with those that have been published previously [40]. SEM was used to analyze the morphology and size of produced ZnONPs. The synthetic ZnONPs had a spherical shape (Figure 3c), and the size, as determined by the graph, was 94.28 nm (Figure 3d).

### 3.3. GOx Immobilization on ZnONPs

ZnONPs with cysteine modifications have GOx covalently bonded to them. The whole process was built upon a two-step mechanism. Cysteine was initially adsorbed onto ZnO using a thiol group, and then the cysteine region was activated using glutaraldehyde. The second phase included using an imine linkage to covalently connect the GOx to the glutaraldehyde carbonyl group. By using cysteine to modify ZnONPs, one carboxyl group (-COOH) and one amino group ($-NH_2$) are formed, remaining electrically neutral, as shown by the chemistry of these two processes. Because it is electrically isolated, this type of support has an advantage over others in that it prevents the unintended adsorption of enzymes on the support. Bezbradica et al.'s investigations [35] also provided an explanation for this phenomenon. They claim that due to the electrostatic connection between the opposing charges of the protein and the support, the charges are vulnerable to undesired deposition of the enzyme, which may block part of the enzyme's catalytic sites [35].

To verify that GOx was immobilized on the altered ZnONPs surface or remained in its original condition after immobilization, FTIR evaluation was carried out. In addition to the amide-I (the peptide stretching pulses of the -C=O group) and amide-II (NH in-plane bending and CN stretching modes of the polypeptide chains), respectively, the FTIR range of native GOx displayed two distinctive peaks at 1648 cm$^{-1}$ and 1530 cm$^{-1}$ (Figure 4, blue curve) [41]. These pairs of peaks at 1648 cm$^{-1}$ and 1530 cm$^{-1}$ were also seen in the FTIR spectra of the GOx/ZnONPs, demonstrating unequivocally that the GOx maintained its active state after being immobilized on ZnONPs [41,42]. Additionally, the FTIR spectra of GOx revealed a distinctive band of the $NH_2$ group in the range of 3000–3500 cm$^{-1}$ [43]. This band has a small upward shift in the FTIR spectra of GOx/ZnONPs. This demonstrated

unequivocally how the NH$_2$ group contributed to the immobilization of GOx on the exterior of ZnONPs by imine linkage.

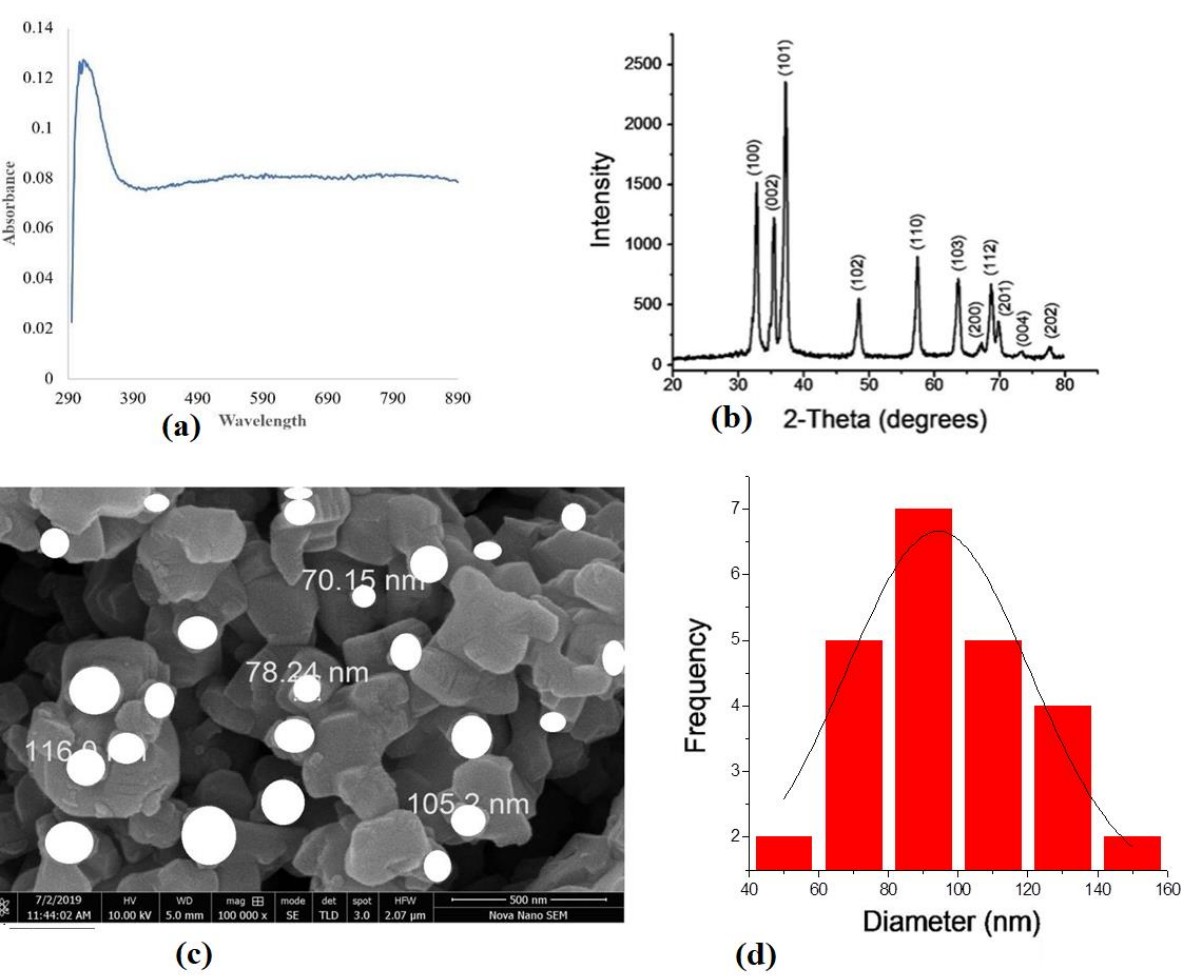

**Figure 3.** (**a**) UV–vis spectrum of ZnONPs; (**b**) XRD spectrum of ZnONPs; (**c,d**) SEM image and histogram of ZnONPs.

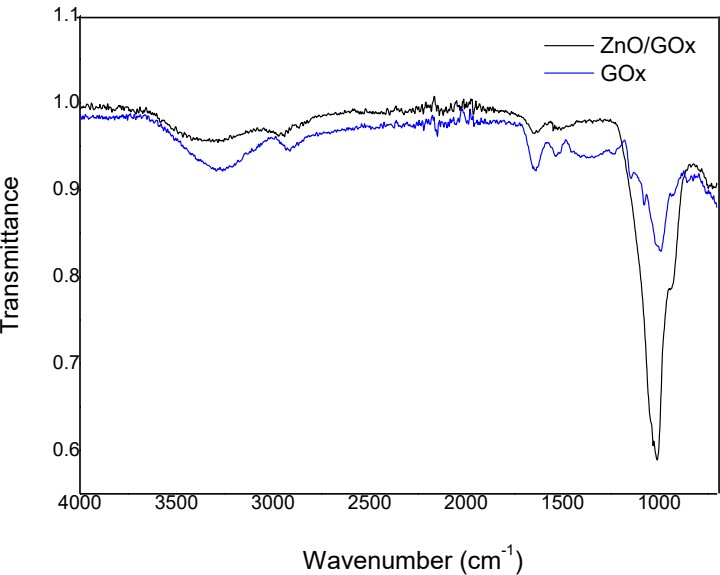

**Figure 4.** FTIR spectra of GOx/ZnONPs (black) and GOx (blue).

### 3.4. GOx/ZnONPs Bioconjugate SEM Imaging

To obtain some knowledge of the morphological features in ZnONPs after GOx immobilization, SEM pictures of GOx/ZnONPs were collected. In comparison to ZnONPs without immobilization, the deposition of GOx on the exterior of ZnONPs demonstrated a propensity to build a more complex pattern of GOx/ZnONPs (Figure 5). These forms of morphological alterations in nanoparticles caused by biomolecules during settlement on their surface have also been documented in research by Kazmi et al. [44].

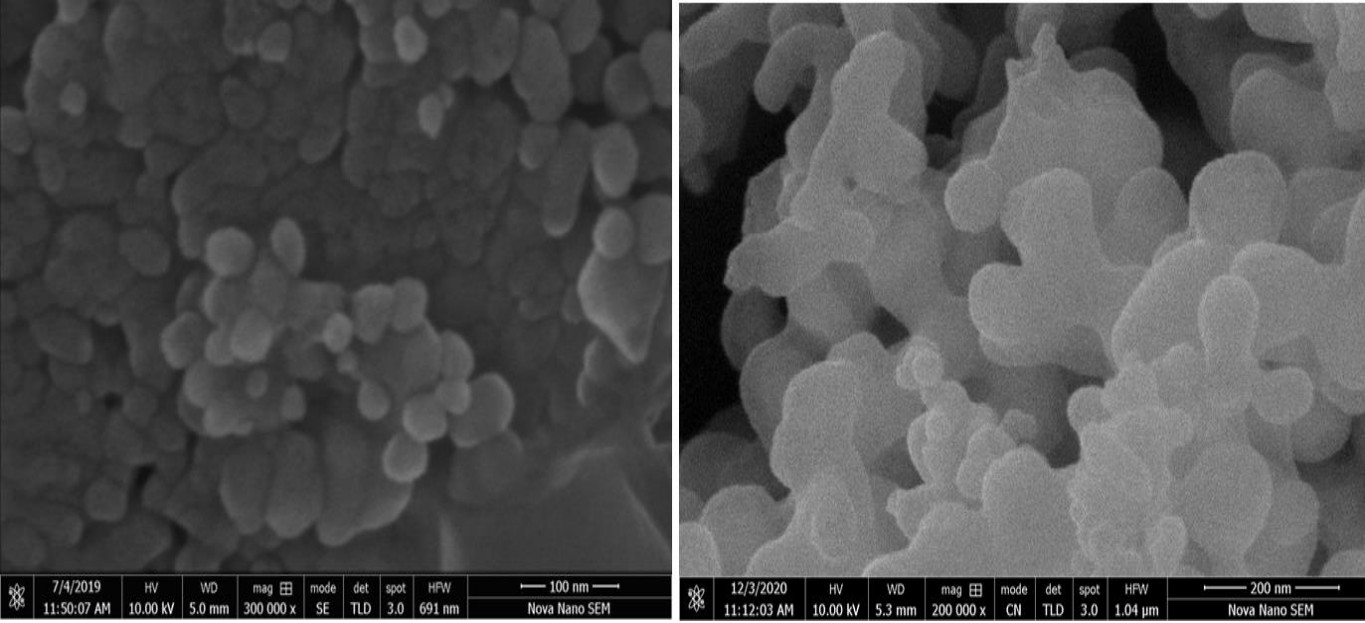

**Figure 5.** SEM images of ZnONPs (**Left**) and GOx/ZnONPs (**Right**).

### 3.5. Analysis of the Free and Adsorbed Enzyme's Function

At operating conditions (pH 7- and 25 degrees Celsius), the catalytic performance of free and adsorbed GOx was determined. It was shown that the unbound enzyme had lower activity ($18.1 \pm 0.33$ U/mL) than the bound enzyme ($23.3 \pm 2.08$ U/mL). For liberated GOx function, a pH of 5.5 was ideal. The deviance from the ideal pH value could be the cause of the decline in liberated enzyme activity. The bound enzyme, however, demonstrated considerably greater activity at pH 7, indicating that it has a broader pH range of action than the liberated enzyme. This could be because the immobilization of an enzyme on a surface changes its microenvironment, which alters the pH range in which the enzyme is active [45].

### 3.6. Impact on Bread's Shelf Life of GOx/ZnONPs Bioconjugate Solution

The GOx/ZnONPs solution was employed in the mixing phase of the bread making procedure to evaluate its impact on the prolonged storage of the bread. Up to 7 days were spent observing the bread. The findings revealed that the treated samples significantly outperformed the control in terms of fungal decay resilience and texture preservation up to the end of the week, while the control was very quickly exposed to spore germination and crust shape degradation after day three (Figure 6). There are several theories as to why GOx/ZnONPs prolong the bread's shelf life. First off, the GOx/ZnONPs bioconjugate functions as both an effective antibacterial and an oxygen sequester [46,47]. Second, we hypothesized that treated samples would produce $H_2O_2$ because of the biocatalytic function of the GOx/ZnONPs bioconjugate. By sealing the cracks in the surface, this film of $H_2O_2$ decreased moisture loss and preserved the integrity of the doughy crust. Finally, the $H_2O_2$ in the samples treated also acted as a potent antiseptic against microorganisms, preventing fungal development in the test samples. The research's conclusions are consistent with past

literature. For instance, Desikan et al. commented on the impact of $H_2O_2$ on the bread's shelf life [48].

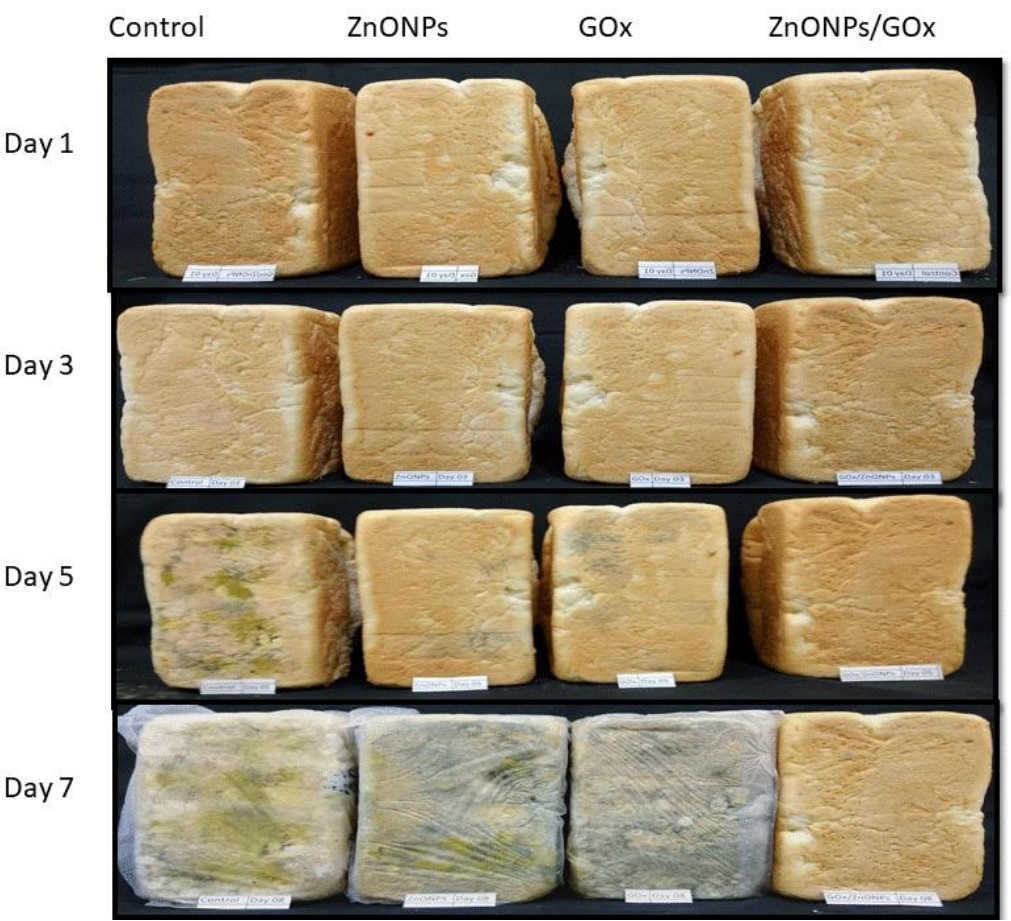

**Figure 6.** Fungus growth showing shelf life of bread.

### 3.7. Bread Quality Parameters

Bread height and volume are important parameters to evaluate its market value. Quality parameters of bread that were studied using application tests were mainly bread height, volume, and microbiological load, especially yeast and mold. Height was highest, i.e., 168 mm with GOx/ZnONPs, 160 mm with GOx, and 155 mm with ZnO NPs, and was lowest with the control sample, as shown in Figures 7 and 8C. GOx/ZnONPs showed their enhanced effect due to their higher activity and stability. A similar effect was seen on the volume of bread, the highest value being 2370 cm$^3$ with GOx/ZnONPs, while it was 2050 mm with GOx, 2010 mm with ZnO NPs, and lowest, i.e., 1825 cm$^3$, with the control sample, as shown in Figures 7 and 8D.

Microbiological load, i.e., yeast and mold, increased with time and reached up to an acceptable limit on the seventh day with GOx/ZnONPs, on the fifth day with GOx and ZnO NPs, and on the fourth day with the control sample, as shown in Figure 8B. It showed that the antimicrobial effect of GOx and ZnO NPs was summed up in GOx/ZnONPs, which ultimately increased the shelf life of bread up to 07 days, which was just 03 days in the control sample.

The effect of GOx/ZnONPs on bread, its shelf life, yeast, and mold count is shown in Figure 8.

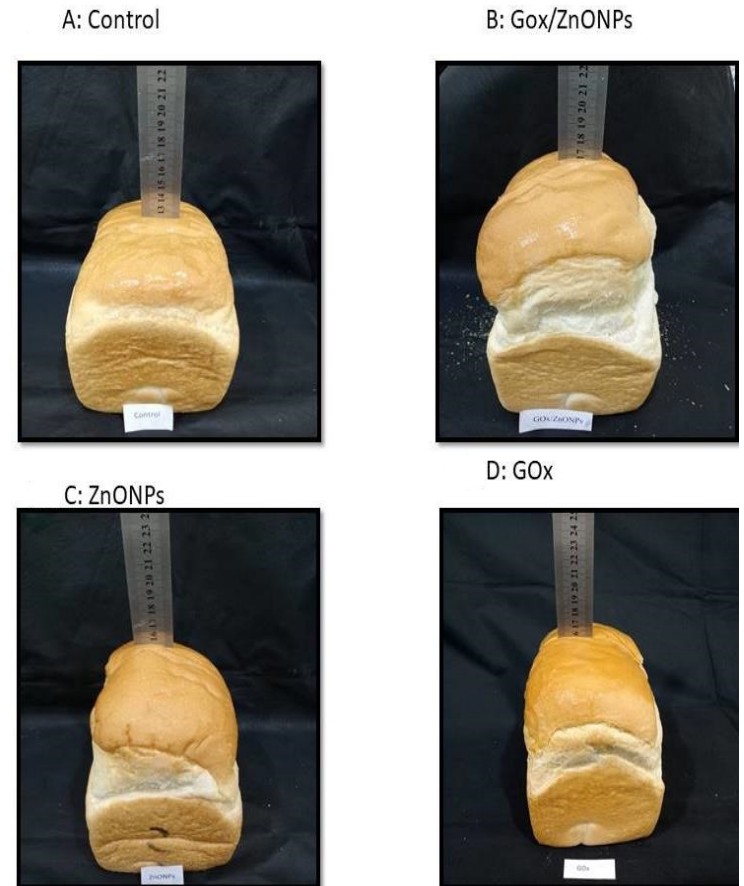

**Figure 7.** Quality parameters of the bread: (**A**) control; (**B**) GOx/ZnONPs; (**C**) ZnONPs; (**D**) GOx.

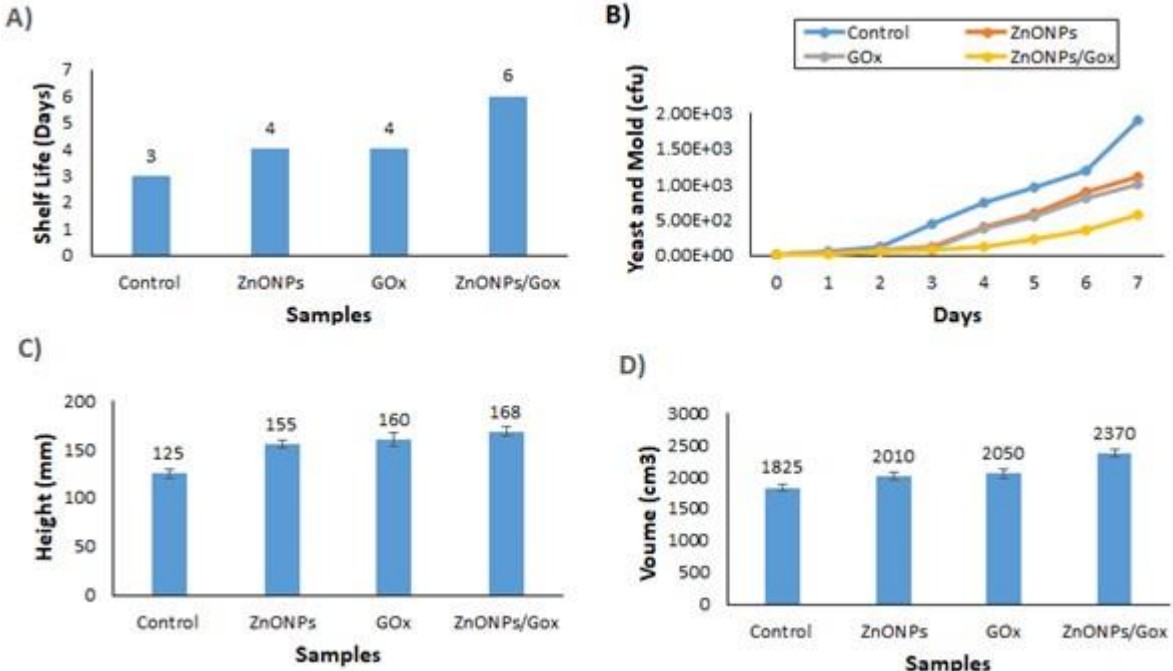

**Figure 8.** Effect of GOx/ZnONPs on bread (**A**) shelf life, (**B**) yeast and mold count, (**C**) height, and (**D**) volume.

### 3.8. Sensory Evaluation of Bread Quality Attributes

Bread quality attributes of softness, fitness, and regularity in crumb texture and whiteness of crumb were given a sensory evaluation against the control. As per the sensory evaluation, all addressed quality attributes were significantly improved in GOx and ZnONPs, with the best results in GOx/ZnONPs (Figure 9).

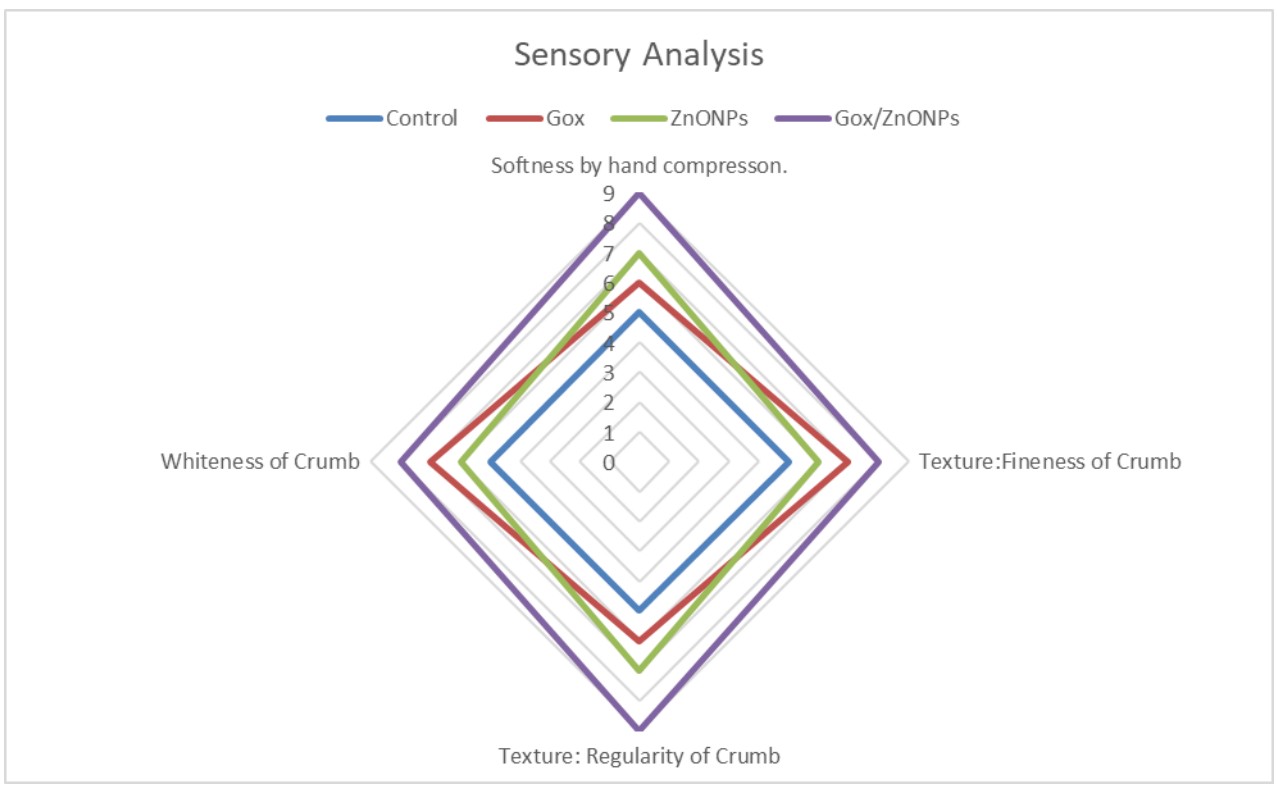

**Figure 9.** Sensory evaluation of bread samples.

### 4. Conclusions

The utilization of GOx immobilized on ZnONPs formed the basis of a unique method for extending the shelf life of bread. Following immobilization of the GOx on ZnONPs, examination showed that the immobilized enzyme was more active than its free equivalents. The GOx/ZnONPs bioconjugate proved effective for treating bread in several ways. First off, the GOx/ZnONPs bioconjugate is an effective oxygen scavenger and antibacterial. This $H_2O_2$ layer also served as a potent disinfectant and guarded the bread from microbial degradation. The biocatalytic activity of this bioconjugate formed a thin coating of $H_2O_2$ that covered the bread. The combination of all the elements created an outstanding framework that significantly extended the shelf life of the bread. The suggested GOx/ZnONPs bioconjugate has excellent industrial potential and may be utilized to extend the shelf life of other food products. To increase the usability and efficacy of the synthetic GOx/ZnONPs bioconjugate spray, more tests are being conducted under various storage environments and on a range of food products.

**Author Contributions:** Conceptualization, J.K., S.K., T.A. and A.S.; methodology, T.A. and A.A.A.; software, A.A.N.; validation, A.A.K.; formal analysis J.K. and A.S.; investigation, S.K.; resources, T.A.; data curation, M.Y.S. and M.T.; writing—original draft, G.G., M.T. and A.A.A.; preparation, A.A.K.; writing—review and editing, M.Y.S., A.V.R. and F.Z.F.; visualization, M.N., S.I.M. and U.A.; supervision, S.K. and T.A.; project administration, S.K.; funding acquisition, S.K. and T.A. All authors have read and agreed to the published version of the manuscript.

**Funding:** This research work was funded by the Higher Education Commission of Pakistan under NRPU 7657.

**Institutional Review Board Statement:** Not applicable.

**Informed Consent Statement:** Not applicable.

**Data Availability Statement:** All major data generated and analyzed in this study are included in this manuscript.

**Acknowledgments:** This work is based upon the work from COST Action18101SOURDOMICS-Sourdough biotechnology network towards novel, healthier and sustainable food and bioprocesses (https://sourdomics.com/; https://www.cost.eu/actions/CA18101/; https://www.cost.eu/actions/CA18101/), where the authors [A.V.R., G.G., and M. T] are [Members] of the Working Groups [7 and 8]. SOURDOMICS supported by COST (European Cooperation in Science and Technology). COST is a funding agency for research and innovation networks. COST Actions help connect research initiatives across Europe and enable scientists to grow their ideas by sharing them with their peers—thus boosting their research, career, and innovation.

**Conflicts of Interest:** The authors declare no conflict of interest.

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
