# Peer review of "Enhancing Bread Quality and Shelf Life via Glucose Oxidase Immobilized on Zinc Oxide Nanoparticles—A Sustainable Approach towards Food Safety"

_sustainability, doi:10.3390/su142114255_

Round 1
Reviewer 1 Report
The manuscript entitled “Enhancing the Bread quality and shelf life via Glucose Oxidase Immobilized on Zinc Oxide Nanoparticles. A sustainable approach towards food safety” describes the immobilization of glucose oxidase in zinc oxide nanoparticles to increase bread quality and shelf life. They showed interesting results related to immobilization process, which is not new, but interesting. The innovation is related to the use of the immobilized enzyme in bread. This should be clearer in the introduction section.
Despite the interesting results of bread quality improvement with the enzyme, the manuscript needs extensive revision in terms of English, organization, standardization, etc. It also needs to improve discussion and answer some important questions.
Many points need correction (but not only those):
Title: “Enhancing the Bread quality and shelf life via…” correction: Enhancing bread quality and…
Line 38: “ Aspergillus Niger-purified enzyme” name of microorganism must be in italics and name of specie begins in lower case letter. Also in lines 74-75.
Line 38 “Aspergillus Niger-purified enzyme was localized on zinc oxide nanoparticles” immobilized?
Line 39: “ZnONPs” what does that mean? It should be clear
Abstract should be revised for better comprehension; Results (in number) should be presented in the abstract. For example “Contrary to control, the use of GOx/ZnONPs significantly 45 improved bread quality, particularly bread height, crumb colour, and volume” which numbers confirm that?
Lines 58-59: “So, the biggest issue for the food industry to serve the customers with best texture with natural aroma of these food products throughout its shelf life.” Something is missing in this sentence. Please revise.
H2O2, KBrO3 the numbers in Chemical formula should be subscript. Correct all
Line 83 “However, it KBrO3 has been banned ...” correct sentence.
Lines 88-90: “But the big problem, its 88 instability that why it has limited industrial applications, therefore going forward to novel target to make it stable. Enzyme immobilization.is such approach.” Please revise sentence
Line 91: BY
Lines 91-92 “Enzyme immobilization, BY physical adsorption or covalent attachment Enzyme is immobilized that makes the enzyme more stable and functional.” Please revise sentence
Lines 105-108: “The usage of zinc oxide nanoparticles (ZnONPs), which have antibacterial capabilities and include zinc, its necessary nutritional supplement, is one of the most popular strategies for immobilizing enzymes. ZnONPs are now reported in the packaged food market as a preferred alternative.” References for those affirmatives should be given. Besides, correction of the sentence is needed.
“The purpose of this 110 research was to investigate how the bioconjugate of GOx/ZnONPs affected the bread's lifespan and quality.” And how are you taking it out of the bread? Some comment should be made about that.
Methodology
“2, 2-Diphenyl-116 1-picrylhydrazyl (DPPH) (Glucose Oxidase refined from Aspergillus Niger, baker's yeast, sucrose, and Oxide Germany’s lysine medium, Novobiocin, carbon base yeast, Glycine, L-Lysine monohydrochloride, Bacteriological agar No. 1.” Revise
“100mL of zinc nitrate with 0.5 strength”. 0.5 strength?
“for 15 minutes to get white ppt” ppt?
“The mixture was subjected allowed to dry...” correction
“Phosphate buffer with 0.1M strength” please do not use “strength”, just 0.1 M is clear enough.
“Aspergillus Niger was cultivated under standardised (standardized) growth conditions to produce the GOx enzyme.” Please use the correct format of microbial name. Please specify growth conditions.
Line 154: use a subtitle for the FTIR analysis.
“The the reaction mixture's absorbance was calculated at 546 nm with a UV-vis spectrophotometer” absorbance was measured, not calculated.
“followed by the addition of a tiny quantity of surfactant.” Even tiny, it should be specified here the amount.
“Then after fermentation at 25 to 27...” 25 to 27 what?
“Volume of bread was measured by seed displacement apparatus” this should be a separate subtitle
“Seed displacement apparatus as shown in Figure 1, was uses for measurement of 194 volume of different bread with.” Revise sentence
Results
Line 220-225: revise sentences. Why this paragraph is necessary? The goal should be set at the end of the introduction and methodology should be clear for the reader to understand assay strategy.
“To verify that GOx was immobilized on the altered ZnONPs surface and that it remained in its original condition after immobilization” to verify IF, not that...
Line 258 and others: “cm-1”. The “-1” value is superscript
Line 279-280: “It was shown that the unbound enzyme had lower activity (18.1 0.33 U/mL) than the bound enzyme (23.3 2.08 U/mL).” These values are confusing. \is it 18.1 or 0.33? 23.3 or 2.08?
Lines 280-284: “For liberated GOx function, a pH of 5.5 was ideal.” ; “The bound enzyme, however, demonstrated considerably greater activity at pH 7, indicating that it has a broader pH range of action than the liberated enzyme.” Where are those results? Just because it has a greater activity at pH7 does not mean it has a broader pH range.
Line 295-300: “Second, we hypothesised (HYPOTHESIZED) that treated samples would produce H2O2 as a consequence of the bio-catalytic function of the GOx/ZnONPs bioconjugate.” “Finally, the H2O2 in the samples treated also acted as a potent antiseptic against the microorganism, preventing fungal development in the test samples “ Why the detection of H2O2 was not performed to confirm this hypothesis? And why the sample with only GOx had fungus growth? Another important question is: the presence of H2O2 in bread would be ok for a consumer? A child? This must be discussed in the manuscript because it impacts in the conclusion of the manuscript.
Figure 7 C and D: standard deviation is important to identify if differences are significant between samples.
“Thirdly, this H2O2 layer also served as a potent disinfectant and guarded the fruit from microbial degradation. Second, the bio catalytic activity of this bio-conjugate formed a thin coating of H2O2 that covers bread.” Remove “thirdly”, “second” from those sentences. These statements cannot be in the conclusion section because they are speculations. They were not proved.
Author Response
Dear Ms. Whitney Wen
Assistant Editor
Please find the attached response to reviewers’ comments on our article entitled “Enhancing the Bread quality and shelf life via Glucose Oxidase Immobilized on Zinc Oxide Nanoparticles. A sustainable approach towards food safety” with manuscript ID sustainability-1963066
Reviewer 1
The manuscript entitled “Enhancing the Bread quality and shelf life via Glucose Oxidase Immobilized on Zinc Oxide Nanoparticles. A sustainable approach towards food safety” describes the immobilization of glucose oxidase in zinc oxide nanoparticles to increase bread quality and shelf life. They showed interesting results related to immobilization process, which is not new, but interesting. The innovation is related to the use of the immobilized enzyme in bread. This should be clearer in the introduction section.
Despite the interesting results of bread quality improvement with the enzyme, the manuscript needs extensive revision in terms of English, organization, standardization, etc. It also needs to improve discussion and answer some important questions.
Many points need correction (but not only those):
Title: “Enhancing the Bread quality and shelf life via…” correction: Enhancing bread quality and…
AR: Thank you very much for your comment it has been changed accordingly in the manuscript. Please see revised manuscript.
Line 38: “ Aspergillus Niger-purified enzyme” name of microorganism must be in italics and name of specie begins in lower case letter. Also in lines 74-75.
AR: Thank you very much for your comment it has been changed accordingly in the manuscript. Please see revised manuscript.
Line 38 “Aspergillus Niger-purified enzyme was localized on zinc oxide nanoparticles” immobilized?
AR: Thank you very much for your comment. It was a mistake, and it has been changed accordingly in the manuscript. Please see revised manuscript.
Line 39: “ZnONPs” what does that mean? It should be clear
AR: Thank you very much for your comment. ZnOPs stands for Zinc nanoparticles it has been changed accordingly in the manuscript. Please see revised manuscript.
Abstract should be revised for better comprehension; Results (in number) should be presented in the abstract. For example “Contrary to control, the use of GOx/ZnONPs significantly 45 improved bread quality, particularly bread height, crumb colour, and volume” which numbers confirm that?.
AR: Thank you very much for your comment. It has been Changed in the manuscript into and values were added in the abstract.
Lines 58-59: “So, the biggest issue for the food industry to serve the customers with best texture with natural aroma of these food products throughout its shelf life.” Something is missing in this sentence. Please revise.
AR: Thank you very much for your comment. It has been Changed in the revised manuscript into bread industry. Please see revised manuscript.
H2O2, KBrO3 the numbers in Chemical formula should be subscript. Correct all
AR: Thank you very much for your comment. It has been Changed in the revised manuscript. Please see revised manuscript.
Line 83 “However, it KBrO3 has been banned ...” correct sentence.
AR: Thank you very much for your comment It has been Changed in the revised manuscript
Lines 88-90: “But the big problem, its 88 instability that why it has limited industrial applications, therefore going forward to novel target to make it stable. Enzyme immobilization.is such approach.” Please revise sentence
AR: Thank you very much for your comment It has been Changed in the revised manuscript
Line 91: BY
Lines 91-92 “Enzyme immobilization, BY physical adsorption or covalent attachment Enzyme is immobilized that makes the enzyme more stable and functional.” Please revise sentence.
AR: Thank you very much for your comment It has been Changed in the revised manuscript
Lines 105-108: “The usage of zinc oxide nanoparticles (ZnONPs), which have antibacterial capabilities and include zinc, its necessary nutritional supplement, is one of the most popular strategies for immobilizing enzymes. ZnONPs are now reported in the packaged food market as a preferred alternative.” References for those affirmatives should be given. Besides, correction of the sentence is needed.
AR: Thank you very much for your comment It has been revised in the revised manuscript
“The purpose of this 110 research was to investigate how the bioconjugate of GOx/ZnONPs affected the bread's lifespan and quality.” And how are you taking it out of the bread? Some comment should be made about that.
AR: Thank you very much for your comment It has been revised as “As bioconjugate of GOx/ZnONPs is within tolerance limit of human body”. Please see revised manuscript.
Methodology
“2, 2-Diphenyl-116 1-picrylhydrazyl (DPPH) (Glucose Oxidase refined from Aspergillus Niger, baker's yeast, sucrose, and Oxide Germany’s lysine medium, Novobiocin, carbon base yeast, Glycine, L-Lysine monohydrochloride, Bacteriological agar No. 1.” Revise
AR: Thank you very much for your comment It has been revised in the revised manuscript.
“100mL of zinc nitrate with 0.5 strength”. 0.5 strength?
AR: Thank you very much for your comment It has been revised in the revised manuscript.
“for 15 minutes to get white ppt” ppt?
AR: Thank you very much for your comment It has been revised in the revised manuscript.
“The mixture was subjected allowed to dry...” correction
AR: Thank you very much for your comment It has been revised in the revised manuscript.
“Phosphate buffer with 0.1M strength” please do not use “strength”, just 0.1 M is clear enough.
AR: Thank you very much for your comment It has been revised in the revised manuscript.
“Aspergillus Niger was cultivated under standardised (standardized) growth conditions to produce the GOx enzyme.” Please use the correct format of microbial name. Please specify growth conditions.
AR: Thank you very much for your comment It has been revised in the revised manuscript.
Line 154: use a subtitle for the FTIR analysis.
AR: Thank you very much for your comment suitable FTIR analysis has been added to the revised manuscript.
“The the reaction mixture's absorbance was calculated at 546 nm with a UV-vis spectrophotometer” absorbance was measured, not calculated.
AR: Thank you very much for your comment It has been revised in the revised manuscript.
“followed by the addition of a tiny quantity of surfactant.” Even tiny, it should be specified here the amount.
AR: Thank you very much for your comment It has been revised in the revised manuscript.
“Then after fermentation at 25 to 27...” 25 to 27 what?
AR: Thank you very much for your comment It has been revised in the revised manuscript.
“Volume of bread was measured by seed displacement apparatus” this should be a separate subtitle
AR: Thank you very much for your comment It has been revised in the revised manuscript.
“Seed displacement apparatus as shown in Figure 1, was uses for measurement of 194 volume of different bread with.” Revise sentence
AR: Thank you very much for your comment It has been revised in the revised manuscript.
Results
Line 220-225: revise sentences. Why this paragraph is necessary? The goal should be set at the end of the introduction and methodology should be clear for the reader to understand assay strategy.
AR: Thank you very much for your comment It has been revised in the revised manuscript.
“To verify that GOx was immobilized on the altered ZnONPs surface and that it remained in its original condition after immobilization” to verify IF, not that...
AR: Thank you very much for your comment It has been revised in the revised manuscript.
Line 258 and others: “cm-1”. The “-1” value is superscript
AR: Thank you very much for your comment It has been revised in the revised manuscript.
Line 279-280: “It was shown that the unbound enzyme had lower activity (18.1 0.33 U/mL) than the bound enzyme (23.3 2.08 U/mL).” These values are confusing. \is it 18.1 or 0.33? 23.3 or 2.08?
AR: Thank you very much for your comment It has been revised in the revised manuscript.
Lines 280-284: “For liberated GOx function, a pH of 5.5 was ideal.” ; “The bound enzyme, however, demonstrated considerably greater activity at pH 7, indicating that it has a broader pH range of action than the liberated enzyme.” Where are those results? Just because it has a greater activity at pH7 does not mean it has a broader pH range.
AR: Thank you very much for your comment It has been revised in the revised manuscript.
Line 295-300: “Second, we hypothesised (HYPOTHESIZED) that treated samples would produce H2O2 as a consequence of the bio-catalytic function of the GOx/ZnONPs bioconjugate.” “Finally, the H2O2 in the samples treated also acted as a potent antiseptic against the microorganism, preventing fungal development in the test samples “ Why the detection of H2O2 was not performed to confirm this hypothesis? And why the sample with only GOx had fungus growth? Another important question is: the presence of H2O2 in bread would be ok for a consumer? A child? This must be discussed in the manuscript because it impacts in the conclusion of the manuscript.
AR: Thank you very much for your comment. As H2O2 produced within the dough so difficult to measure so accurately. Only GOx has fungus growth but has more shelf life than control one. However GOx/ZnONPs bioconjugate due to its stability and more activity produce more H2O2 resulting in extended shelf life. Finally, in backing process at 170°C H2O2 decomposed so unavailable for consumer.
Figure 7 C and D: standard deviation is important to identify if differences are significant between samples.
AR: Thank you very much for your comment. Standard deviation has been added to figure 7C and D in the revised manuscript.
“Thirdly, this H2O2 layer also served as a potent disinfectant and guarded the fruit from microbial degradation. Second, the bio catalytic activity of this bio-conjugate formed a thin coating of H2O2 that covers bread.” Remove “thirdly”, “second” from those sentences. These statements cannot be in the conclusion section because they are speculations. They were not proved.
AR: Thank you very much for your comment. It has been changed accordingly.
Regards
Dr. Tariq Aziz (Postdoc, PhD)
School of Food & Biological Engineering
Jiangsu University China

Reviewer 2 Report
The article entitled “Enhancing the Bread quality and shelf life via Glucose Oxidase Immobilized on Zinc Oxide Nanoparticles. A sustainable approach towards food safety” is a simple study applying nanoparticles in bread. The topic is interesting, but the story is not complete. As such"
1. Please explain why this research is strongly correlated with global sustainability issue.
2. There is only one samples which was characterized in the nanoparticles. There was no control and no optimization process. If this part of study has been conducted previously, please mention clearly in the method
3. Schematic diagram of making the nanoparticles will help the readers to understand.
4. In the application of nanoparticles in food, sensory aspect must be taken into account. However, the is no sensory analysis in this research.
5. The effect of the nanoparticles on the quality of bread mut be further elaborated.
Based on these comments, I would like to recommend major revision to this article before being accepted for publication in this journal.
Author Response
Dear Ms. Whitney Wen
Assistant Editor
Please find the attached response to reviewers’ comments on our article entitled “Enhancing the Bread quality and shelf life via Glucose Oxidase Immobilized on Zinc Oxide Nanoparticles. A sustainable approach towards food safety” with manuscript ID sustainability-1963066
Reviewer 2
The article entitled “Enhancing the Bread quality and shelf life via Glucose Oxidase Immobilized on Zinc Oxide Nanoparticles. A sustainable approach towards food safety” is a simple study applying nanoparticles in bread. The topic is interesting, but the story is not complete. As such"
- Please explain why this research is strongly correlated with global sustainability issue.
Thank you very much for your comment. Food losses and waste are associated with inefficient use of agricultural land, water and other resources and agricultural raw materials. The distribution of food wastage along the food supply chain is linked to the level of economic development of countries and regions. Developing countries mainly suffer from production and post-harvest handling and storage losses, while in highly developed countries or regions food is mainly wasted at those stages in the food chain where the consumer plays an active role: in distribution and retail, in restaurants and households [Gustavsson et al 2011]. In households in particular, the scale of food waste is strongly correlated with the level of gross domestic product (GDP) [Xue et al 2017]. Solely the negative effects of food losses and waste, in environmental, economic, and social terms, justify the need to intensify research in this area and then undertake preventive measures. This approach also applies to the production of bread and other bakery and confectionary products, even if the research so far shows a small scale of the problem. Nowadays, in the face of the global health crisis related to COVID-19 pandemic, it is of tremendous importance when it has become clear that it is not possible to achieve the second Sustainable Development Goal (SDG) of the United Nations agenda. The SDG 2 aims to achieve “Zero hunger” by 2030 [FoodDrinkEurope Every Crumb Counts; Joint Food Waste Declaration: Brussels, Belgium, 2013 ].
To overcome these issues, nanotechnology is a key advanced technology enabling contribution, development, and sustainable impact on food, medicine, and agriculture sectors. Nanomaterials have potential to lead qualitative and quantitative production of healthier, safer, and high-quality functional foods which are perishable or semi-perishable in nature. Nanotechnologies are superior than conventional food processing technologies with increased shelf life of food products, preventing contamination, and production of enhanced food quality. According to (Cano et al 2018), The release of silver ions helps to reduce microbial load with sustainable development of various aseptic food containers and antimicrobial surfaces, providing active packaging food systems with promising quality. Very low amount of silver ions (10–100 mg Ag t/kg) is required to achieve biocidal efects using in water or low bufered systems. Interestingly, the antimicrobial activity of silver decreases rapidly in the presence of proteins in food system, and hence, the silver amount required was 50–100 mg Agt kg−1 in realistic food applications. The properties and behavior of colloidal particles are important to design foods which are safer and healthier with improved quality and sustainability. In short nanotechnology plays a major role in the food sector through the quality food production ends with advanced processing, packaging, and long-term storage, provided enormous growth in food industry through enhancement in food quality by improving its flavor and texture.
Some of the following references have been highlighted for further clarification.
A.I. Cano, A., C. González-martínez, Silver composite materials and food packaging. In Composites Materials for Food Packaging (Wiley, 2018)), pp. 123–151. https://doi. org/10.1002/9781119160243.ch3
- Naoto, O. Hiroshi, N. Mitsutoshi, Mitsutoshi, micro- and nanotechnology for food processing. (Food safety series) resource: engineering and technology for a sustainable. World. Am. Soc. Agric. Eng. 16, 19 (2009)
S.M. Rodrigues, P. Demokritou, N. Dokoozlian, C.O. Hendren, D.B. Karn et al., Environmental science nanotechnology for sustainable food production: promising opportunities and scientifc challenges. Environ. Sci. Nano 1, 767–781 (2017). https://doi.org/10.1039/c6en00573j
K.P. Chandrika, A. Singh, M.K. Tumma, P. Yadav, Nanotechnology prospects and constraints in agriculture, in Environmental chemistry for a sustainable world, vol. 14, ed. by N. Dasgupta, S. Ranjan, E. Lichtfouse (Springer, New York, 2018)
Lipinski, B.; Hanson, C.; Waite, R.; Searchinger, T.; Lomax, J.; Kitinoja, L. Reducing food loss and waste. Working Paper, Installment 2 of Creating a Sustainable Food Future. Washington, DC: World Resources Institute. 2013. Available online: http://www.worldresourcesreport.org. (accessed on 23 November 2020).
Gustavsson, J.; Cederberg, C.; Sonesson, U. Global Food Losses and Food Waste: Extent, Causes and Prevention; Study Conducted for the International Congress Save Food! At Interpack 2011, 16–17 May, Düsseldorf, Germany; Food and Agriculture Organization of the United Nations: Rome, Italy, 2011; ISBN 978-92-5-107205-9.
Xue, L.; Liu, G.; Parfitt, J.; Liu, X.; Van Herpen, E.; Stenmarck, Å.; O’Connor, C.; Östergren, K.; Cheng, S. Missing Food, Missing Data? A critical review of global food losses and food waste data. Environ. Sci. Technol. 2017, 51, 6618–6633.
FoodDrinkEurope Every Crumb Counts; Joint Food Waste Declaration: Brussels, Belgium, 2013; Available online: https:// everycrumbcounts.eu/ (accessed on 23 November 2020)
Gory ´nska-Goldmann, E.;Gazdecki, M.; Rejman, K.;Kobus-Cisowska, J.; Łaba, S.; Łaba, R. How to Prevent Bread Losses in the Baking and Confectionery Industry?—Measurement, Causes, Management and Prevention. Agriculture 2021, 11, 19. https:// doi.org/10.3390/agriculture11010019.
- There is only one samples which was characterized in the nanoparticles. There was no control and no optimization process. If this part of study has been conducted previously, please mention clearly in the method.
AR; Thank you very much for your comment. It has been revised in the manuscript please check revised manuscript.
- Schematic diagram of making the nanoparticles will help the readers to understand.
AR: Thank you very much for your comment. A schematic diagram has been added to the revised manuscript. Please see revised manuscript.
- In the application of nanoparticles in food, sensory aspect must be taken into account. However, the is no sensory analysis in this research.
AR: Thank you very much for your comment. Sensory evaluation of bread quality attributes has been added to the manuscript please check the revised manuscript.
- The effect of the nanoparticles on the quality of bread mut be further elaborated.
AR: Thank you very much for your comment. It has been added to the results and discussion section in the revised manuscript. Please see revied manuscript.
Based on these comments, I would like to recommend major revision to this article before being accepted for publication in this journal.
Regards
Dr. Tariq Aziz (Postdoc, PhD)
School of Food & Biological Engineering
Jiangsu University, China

Reviewer 3 Report
I recommend the research paper entitled "Enhancing the Bread quality and shelf life via Glucose Oxidase 2 Immobilized on Zinc Oxide Nanoparticles. A sustainable approach towards food safety" for publication in its present form. The authors have well designed their study with appropriate literature while the figures clearly present the overall results.
Author Response
Dear Ms. Whitney Wen
Assistant Editor
Please find the attached response to reviewers’ comments on our article entitled “Enhancing the Bread quality and shelf life via Glucose Oxidase Immobilized on Zinc Oxide Nanoparticles. A sustainable approach towards food safety” with manuscript ID sustainability-1963066
Reviewer 3
I recommend the research paper entitled "Enhancing the Bread quality and shelf life via Glucose Oxidase 2 Immobilized on Zinc Oxide Nanoparticles. A sustainable approach towards food safety" for publication in its present form. The authors have well designed their study with appropriate literature while the figures clearly present the overall results.
AR: Thank you very much for your appreciation of our work.
Regards
Dr. Tariq Aziz (Postdoc, PhD)
School of Food & Biological Engineering
Jiangsu University China

Round 2
Reviewer 1 Report
The manuscript has been revised and can now be accepted for publication.
Reviewer 2 Report
The authors have improved the manuscript based on on the comments.